# Structural Parameters on the Effective Transport Properties of Carbon Cloth Gas Diffusion Layers: Random Walk Simulations

**DOI:** 10.3390/nano15040259

**Published:** 2025-02-09

**Authors:** Qingrong Jia, Hao Wang, Guogang Yang

**Affiliations:** Marine Engineering College, Dalian Maritime University, Dalian 116026, China

**Keywords:** carbon cloth gas diffusion layer, transport property, random walk simulation

## Abstract

One of the key challenges in optimizing the transfer characteristics of carbon cloth gas diffusion layers (GDLs) is accurately evaluating their effective transport properties. In this work, a stochastic reconstruction algorithm based on structural parameters was developed to generate virtual carbon cloth GDLs with varying porosities, carbon fiber diameters, and length-to-thickness ratios. A pore-scale random walk model was also developed to predict the permeability, tortuosity, and effective diffusivity of the GDLs with well-validated accuracy. The results show that higher porosity and larger carbon fiber diameters enhance the diffusion and transfer of oxygen through the GDL, and when the length-to-thickness ratio exceeds 2, the permeability stabilizes. The model developed in this study offers the advantages of low computational cost, accurate representation of the material’s microstructure, and broad applicability, making it a powerful and convenient tool for predicting the transport properties of porous media.

## 1. Introduction

The carbon cloth gas diffusion layer (GDL) is a crucial component of fuel cells, made from conductive carbon fibers with a porous structure and outstanding electrical conductivity [1]. Its primary functions include supporting the catalytic layer, ensuring the uniform distribution of reactant gases, facilitating the removal of generated water and heat, and providing an efficient pathway for electron transfer, among other diverse roles [2]. With its high electrical conductivity, excellent gas permeability, strong corrosion resistance, and notable mechanical strength, the carbon cloth GDL significantly enhances the performance and operational stability of fuel cells. Through careful design optimization, the GDL not only enables efficient mass transfer within the electrode structure but also contributes to improving the overall durability of the system, highlighting its importance in advancing fuel cell technology [3]. The microstructural characteristics of the carbon cloth GDL, such as porosity and carbon fiber diameter, greatly influence its transport properties. These properties include permeability, tortuosity, and effective diffusivity, all of which are critical for performance [4,5]. Optimizing these microstructural parameters can effectively enhance the gas transport efficiency and water management capabilities of fuel cells. However, significant gaps remain in current research regarding the relationship between the microstructure of carbon cloth GDL and its performance, particularly in areas such as three-dimensional structural reconstruction and performance prediction. Consequently, understanding how to precisely control the microstructure of the carbon cloth GDL to improve its performance has emerged as a key scientific challenge in the field of fuel cells.

Carbon cloth GDLs are widely used in PEMFCs due to their high electrical conductivity, excellent mechanical strength, and adaptability. In recent years, extensive studies have been conducted on the relationship between the microstructure and performance of carbon cloth GDLs through both experimental characterization and numerical simulation. Experimental characterization is considered a crucial method for studying the microstructure of GDLs. The electrical conductivity and microstructural changes of carbon cloth GDLs under varying compression conditions were investigated by Mariani et al. [6]. It was found that, as compressive pressure increased, significant changes in electrical conductivity and the microstructure of the carbon cloth were observed. Specifically, when the compressive pressure was increased to 2.0 MPa, fiber cracks were caused, leading to a reduction in electrical conductivity. The feasibility of using PAN-based carbon cloth as a gas diffusion layer for PEMFCs was examined by Liu et al. [7]. It was demonstrated that this type of carbon cloth possesses excellent compressibility and elasticity, making it suitable for application as a GDL. After being coated with a microporous layer (MPL), its discharge curves were found to be comparable to those of conventional carbon papers. The wettability and evaporation behaviors of a carbon cloth-type GDL in passive direct alcohol fuel cells were analyzed by Nagy et al. [8]. It was revealed that ethanol showed better spreading characteristics on the carbon cloth compared to methanol, especially on the microporous side. Furthermore, it was observed that water evaporation on the carbon cloth occurred in two distinct phases: a constant contact angle mode and a constant contact radius mode. The transition between these phases was noted at approximately 0.65 relative evaporation time. The effects of different concentrations of hydrophobic agents on the performance of carbon cloth GDLs were explored by Qiu et al. [9]. It was demonstrated that increasing the concentration of the hydrophobic agent led to an increase in GDL thickness, air permeability, and surface resistivity, whereas electrical conductivity was reduced. At an operating temperature of 80 °C, it was shown that a GDL containing 5% FEP achieved the best performance in terms of battery efficiency.

Through these studies, valuable insights into the relationship between the microstructural characteristics and performance of carbon cloth GDLs have been provided, contributing to the further optimization of fuel cell applications. Although experimental methods can yield accurate and detailed information, they are often time-consuming and costly. Consequently, numerical simulation has emerged as an important tool to guide the design of PEMFCs. Researchers have focused on developing macro-scale models of PEMFCs to gain a deeper understanding of internal heat and mass transfer mechanisms, thereby optimizing the design of fuel cell systems and components. Aquah et al. [2] conducted a detailed investigation into the hydrodynamic properties of woven carbon cloth GDLs using computational fluid dynamic (CFD) simulations within the OpenFOAM framework. It was revealed that the in-plane permeability of the carbon cloth GDL is significantly lower than its through-plane permeability. This finding emphasizes the critical role of fiber alignment and orientation in determining the permeability characteristics of the GDL. Additionally, the spatial heterogeneity of porosity and permeability within the GDL, particularly in the Sigracet SGL 25 BA GDL, was highlighted by their research. Kopanidis et al. [10] examined pore-scale thermal and mass transfer phenomena within a segment of the PEM cathode channel and carbon cloth GDL. A direct 3D microscale model was utilized to investigate local heat and fluid flow parameters, as well as their effects on water vapor condensation, which can lead to flooding. The simulation results offered insights into how these phenomena affect the overall performance and durability of the system under specific operating conditions. In a study by García-Salaberri [11], a composite continuum network model was developed for carbon cloth GDLs. This model combined macroscopic control volumes with microscopic pore network characteristics to analyze local effective transport properties. It was designed to simulate both diffusion and convection across multiple scales, with the goal of optimizing the overall performance of PEMFCs. These studies collectively provide a comprehensive understanding of the fluid dynamics, thermal behavior, and water management processes within carbon cloth GDLs. They underscore the importance of factors such as fiber orientation, porosity distribution, and multiphase flow interactions in enhancing the performance of PEMFCs. By leveraging these findings, future improvements in the design and efficiency of fuel cells can be achieved.

In the macro-scale modeling of a PEM fuel cell equipped with a carbon cloth GDL, the accurate prediction of macroscopic transport parameters critically depends on prior pore-scale investigations. The increasing adoption of stochastic reconstruction algorithms in GDL microstructure simulations has advanced this field. This approach involves generating three-dimensional models with varying porosities and fiber architectures randomly, which are then coupled with fluid dynamics models to predict gas diffusion behaviors within these complex microstructures. Rama et al. [12] analyzed the effects of compression on the structural integrity and transport properties of carbon cloth GDLs. Digital reconstruction techniques were employed to observe structural deformations under various compression loads, whereas anisotropic permeability was modeled using the lattice Boltzmann method (LBM). Their findings established correlations between compressive pressure, permeability, and porosity, elucidating the impact of compression on transport performance. Moreover, a three-dimensional digital model of the carbon cloth GDL was developed using X-ray computed tomography (XCT) combined with single-phase LBM, enabling the simulation of in-plane and through-plane permeabilities [13]. Molaeimanesh et al. [14] explored the influence of microstructure on the cathode performance of PEMFCs. By reconstructing three distinct microstructures of carbon cloth GDLs, in which carbon fiber bundles were modeled with sinusoidal forms, gas-phase flow within these GDLs was simulated using LBM. The study revealed that an increase in fiber content within the carbon cloth bundles significantly impeded oxygen diffusion toward the catalyst layer, leading to a reduction in average current density. Interestingly, the in-plane compactness of the carbon cloth microstructure was found to have a negligible effect on current density or species distribution. Gao et al. [15] performed comprehensive studies on stochastic reconstruction and transport predictions for carbon cloth GDLs. A novel method for constructing 3D microstructures based on real material data was proposed, incorporating image-based characterization to quantify microstructural features, such as porosity, pore size distribution, and cavity morphology. Furthermore, a stochastic reconstruction algorithm was developed to generate anisotropic 3D microstructural models. The effectiveness of these methodologies was validated through classical simulation predictions and detailed evaluations of transport characteristics. Gao et al. also investigated the influence of sample size on the transport properties of carbon paper and carbon cloth GDLs, determining the representative elementary volume (REV) [16], a critical parameter for ensuring the accuracy of macro-scale transport predictions. These studies underscore the pivotal role of pore-scale modeling and stochastic reconstruction in improving the understanding of GDL microstructure–performance relationships. The integration of advanced digital reconstruction techniques with transport simulations enables the precise prediction of permeability, porosity, and other key transport parameters, offering invaluable insights for the optimization of PEM fuel cell systems.

Using stochastic reconstruction algorithms or XCT technology, the three-dimensional structural information of carbon cloth GDLs can be effectively obtained. When combined with the LBM, the fluid transport characteristics within porous media can be accurately predicted. This approach provides robust support for evaluating the transport performance of carbon cloth GDLs with precision. Furthermore, the random walk algorithm has emerged as a widely utilized tool for simulating particle diffusion processes in porous materials, enabling the efficient computation of key parameters such as permeability. This method is notable for its fixed computational cost, independence from frequency, and short execution times, showcasing exceptional performance in predicting the parameters of porous media. Bahadori et al. [17] introduced a mesh-free Monte Carlo method to solve two-dimensional transient heat conduction problems in composite materials with temperature-dependent thermal conductivity. Defrenne et al. [18] applied random walk methods within XCT-generated structures to investigate the dual-phase moisture conductivity of fibrous materials, combining this approach with LBM to predict fluid transport properties in porous media. Wang et al. [19] employed random walk theory to calculate the effective oxygen diffusivity in porous battery electrodes, validating their findings through tomographic techniques. Similarly, Mao et al. [20] developed an efficient numerical solver that integrated finite difference and random walk models, specifically tailored for simulating strongly inhomogeneous diffusion in polycrystalline structures with high accuracy. These studies underscore the versatility and effectiveness of random walk algorithms in modeling complex transport phenomena. By leveraging advanced numerical techniques, such as random walk methods and their integration with XCT and LBM, significant strides have been made in accurately characterizing and predicting the transport properties of porous materials. This highlights their indispensable role in advancing the design and optimization of porous media across various applications.

In this study, a stochastic reconstruction model was developed to generate virtual three-dimensional structures of carbon cloth GDLs with adjustable structural parameters. The model enabled the creation of carbon cloth GDLs with varying porosities, fiber diameters, and length-to-thickness ratios (*LTRs*). Additionally, a random walk model was established to simulate the transport properties of carbon cloth GDL samples, including permeability, tortuosity, and effective diffusivity. The findings of this study provide more accurate predictions of the transport parameters of carbon cloth GDLs, offering critical support for the development of full-scale PEMFC models. By integrating these advanced simulation techniques, the study enhances the understanding of the relationship between the microstructure and transport performance in carbon cloth GDLs and contributes to optimizing their design for improved fuel cell performance and durability.

## 2. Numerical Methods

### 2.1. Microscopic Stochastic Reconstruction Algorithm for Carbon Cloth GDLs

Based on reported work, carbon cloth is woven from multiple carbon bundles, each composed of numerous carbon fibers. To enhance the feasibility of the algorithm, the following assumptions were made [16]:(i).The representative unit of the carbon cloth GDL is divided into two pairs of mutually orthogonal bundles.(ii).Each bundle has an elliptical cross-section.(iii).A bundle consists of many carbon fibers that are randomly distributed.(iv).Carbon fibers are considered cylindrical with a sinusoidal straight direction.

By controlling the basic parameters—the fiber radius (*r*) and the minimum distance (*d*) between the two nearest fibers within the bundle—reconstruction of the carbon cloth was achieved.

For the voxel coordinates *P*(*x,y,z*) contained within the *i*-th carbon fiber, the following conditions need to be satisfied:(1)$${\left(\frac{z-z_{i}-A\sin\left(\frac{2\pi x}{L}\right)}{r}\right)}^{2}+{\left(\frac{y-y_{i}}{r}\right)}^{2}\leq 1$$
where, *A* represents the amplitude of the sinusoidal axis of the carbon fiber, *L* is the wavelength, and *z_i_
*and *y_i_* represent the coordinates of the center of the carbon fiber in the *x* = 0 plane. By repeatedly generating non-overlapping carbon fibers, a single carbon fiber bundle can be obtained. Other orthogonal carbon fiber bundles can be generated through coordinate transformations. By adjusting the number of carbon fibers within a single bundle, the porosity can be controlled.

### 2.2. Random Walk Algorithm

The random walk algorithm is used to simulate the process of random changes in system states. In simple terms, a random walk describes a trajectory where a particle starts from an initial position and, at each step, randomly chooses a direction to move forward based on certain probabilities. The transition from the current position to the next is entirely governed by a probability distribution. In this study, the following widely adopted assumptions had to be declared: (1) the transition rules are equiprobable, allowing particles to randomly select directions to move within a three-dimensional space and (2) the interactions between the gas and the medium surface, as well as the interactions among gas molecules, are neglected, and only the molecular diffusion in free space is considered [17,18,19,20]. If the next position was occupied by a solid, the particle remained stationary for that time step. For each case study, 10^5^ random walkers were placed within the pore space. After 10^5^ movements, the mean squared displacement of the random walkers was calculated as follows [21]:(2)$$\langle r^{2}\left(t\right)\rangle =\frac{1}{n}\sum_{i=1}^{n}\left\{{\left[x_{i}\left(t\right)-x_{i}\left(0\right)\right]}^{2}+{\left[y_{i}\left(t\right)-y_{i}\left(0\right)\right]}^{2}+{\left[z_{i}\left(t\right)-z_{i}\left(0\right)\right]}^{2}\right\}$$
where, *n* represents the number of random walkers. The effective diffusivity, tortuosity, and permeability of the porous medium was calculated as follows:(3)$$D_{eff}=\varepsilon \frac{d{\langle r^{2}\left(t\right)\rangle }_{p}}{6dt}$$(4)$$\tau =\frac{\frac{d\langle r^{2}\left(t\right)\rangle }{dt}}{\frac{d{\langle r^{2}\left(t\right)\rangle }_{p}}{dt}}$$(5)$$K=\frac{\varepsilon }{6\tau {\left(\frac{S}{V}\right)}_{p}^{2}}$$
where, *ε* represents the porosity and *S*/*V* is the surface-to-volume ratio of the porous structure.

### 2.3. Model Validation

To validate the applicability of the random walk model in predicting the transport properties of porous media, the effective diffusivity of a structure with a simple cubic stacking of spheres in a 200 × 200 × 200 space was calculated. The radius of the spheres was gradually reduced to generate a series of porous structures and determine their effective diffusivity. Figure 1 illustrates the variation of the effective diffusivity with porosity, and the inset shows the porous structure for different sphere diameters. Comparing the calculated results with the empirical formulas of Bruggeman et al. [22] and Neale et al. [23] the transport properties obtained by the random walk model are in good agreement with the results of the studies carried out by the previous authors.

## 3. Results and Discussion

### 3.1. The Effect of Porosity

Porosity is a key parameter affecting the transport properties of carbon cloth GDLs, and the reported porosity range of carbon cloth GDLs is 0.7–0.87 [14,24]. Although there have been a number of studies analyzing the effect of porosity on the transport parameters of porous media, no studies have fully explored the effect of porosity on the transport parameters of carbon cloth GDLs. In this study, based on the carbon cloth GDL stochastic reconstruction model developed in Section 2.1, 300 carbon cloth GDL samples were randomly generated within the porosity range of 0.7–0.9, with GDL dimensions of 400 × 400 × 200 μm^3^ and a carbon fiber diameter of 7 μm. Three virtual structures were selected to be displayed in Figure 2. The developed stochastic reconstruction model of the carbon cloth GDL accurately represents the structure of carbon fiber bundles orthogonal to each other. As porosity increases, the number of carbon fibers in the carbon fiber bundles decreases, and the carbon cloth GDL becomes more loose, which favors gas and liquid water transport but reduces mechanical properties.

Figure 3 presents the transport property calculation results for 300 samples of carbon cloth GDLs. The calculated effective diffusivity, permeability, and tortuosity were compared with the relations previously proposed for porous media transport properties, as listed in Table 1. It can be observed that the transport properties of carbon cloth GDLs have strong correlations with porosity, exhibiting very clear patterns. Since previous studies were not specifically conducted on carbon cloth GDLs, the prediction accuracy fluctuates due to differences in microstructural characteristics [22,23,25]. For effective diffusivity, shown in Figure 3a, the correlations of Bruggeman et al. [22] and Neale et al. [23] overestimate the diffusivity, whereas that of Nam et al. [25] underestimates it. The permeability of carbon cloth GDLs increases with increasing porosity, and the tortuosity decreases accordingly. GDLs with higher porosity have simpler transport paths and are more conducive to gas passage. Permeability has a higher consistency with the Nabovati relation [26], but there are still some errors. Therefore, we adopted a modified Kozeny–Carman relation to fit the permeability results, as presented in Equation (7), which provides more accurate fitting results. As the porosity increases, the pore path becomes simpler and closer, so the tortuosity curve is gentler. Regarding the transport parameters of carbon cloth GDLs, this study re-fits the effective diffusivity, permeability, and tortuosity. The resulting relationships provide higher prediction accuracy for the transport parameters of carbon cloth GDLs and can support the establishment of more accurate full-scale fuel cell models. They are as follows:(6)$$D_{eff}=\varepsilon^{2.03},{\;R}^{2}=0.9899$$(7)$$K=0.4957\frac{\varepsilon^{3}}{{\left(1-\varepsilon \right)}^{2}},{\;R}^{2}=0.9989$$(8)$$\tau =\varepsilon^{-1.0547},{\;R}^{2}=0.9587$$

### 3.2. The Effect of Fiber Diameter

The diameter of carbon fibers has a significant impact on the pore structure and gas permeability characteristics of the carbon cloth GDLs. By adjusting the carbon fiber diameter, the porosity and permeability of the GDL can be optimized, thereby enhancing the overall performance of the fuel cell. The carbon fiber diameter is a crucial parameter that influences the macroscopic characteristics of the model. Precise control of the carbon fiber diameter allows for a more accurate simulation of the actual GDL performance, enabling a better understanding of the effects of different carbon fiber diameters on GDL properties, and it is valuable for guiding material selection and process design in practical production. Based on the stochastic reconstruction method, 300 carbon cloth GDL samples were generated with a porosity of 0.78 [14,24], which is the classic value for GDL porosity, and random carbon fiber diameters ranging from 6 to 12 microns. Figure 4 displays the three-dimensional structures of the carbon cloth GDL with different carbon fiber diameters. It is evident that as the carbon fiber diameter increases, the number of fibers decreases, but the structural characteristics of the carbon cloth GDL are still preserved.

Figure 5 illustrates the transport properties of carbon cloth GDLs with varying carbon fiber diameters. Although the porosity remains similar, the number of carbon fibers varies with diameter, modifying the pore size distribution and inevitably altering the transport properties. Figure 5a presents effective diffusivity as a function of carbon fiber diameter, showing an increasing trend with diameter, along with a pronounced discrete distribution. The effective diffusivity relations proposed by Bruggeman et al. [22], Neale et al. [23], and Nam et al [25]. do not account for carbon fiber diameter; thus, the variation with diameter is not significant. The fluctuations in the curve are due to slight differences in porosity. Based on the modified Bruggeman relation, a scaling factor related to the carbon fiber diameter was introduced, and the fitted correlation is given by Equation (9). As shown in Figure 5a, the predictive relation proposed in this study, which considers the carbon fiber diameter, shows a stronger correlation with the results obtained from the random walk model. Figure 5b shows permeability as a function of carbon fiber diameter. As listed in Table 1, previous scholars’ permeability relations considered the carbon fiber radius, where permeability is proportional to *r*^2^. This study provides a modified K–C relation for carbon cloth GDLs, given in Equation (10), with an R^2^ value of 0.9965, confirming its superior predictive accuracy. Figure 5c shows tortuosity as a function of carbon fiber diameter. The previously proposed tortuosity relations did not account for the effect of carbon fiber diameter. However, as shown in Figure 4, the variation in carbon fiber diameter alters the tortuosity of the pores within the carbon cloth GDL. Within the considered range of carbon fiber diameters, tortuosity decreases almost linearly, indicating that larger carbon fiber diameters result in simpler fluid flow paths. This study proposes a tortuosity prediction relation that accounts for the carbon fiber diameter, as given in Equation (11), demonstrating a significantly stronger correlation compared to previous studies.(9)$$D_{eff}=\left(0.0094d+0.8228\right)\varepsilon^{1.5448}$$(10)$$K=\frac{d^{2}}{99.65}\frac{\varepsilon^{3}}{{\left(1-\varepsilon \right)}^{2}}$$(11)$$\tau =\varepsilon^{0.0416d-1.3146}$$

### 3.3. The Effect of the Length-to-Thickness Ratio

As shown in Figure 2 and Figure 4, the carbon cloth GDL was woven from carbon fiber bundles. In the stochastic reconstruction algorithm, the individual carbon fiber curves follow a sine distribution. Therefore, the wavelength of the sine curve, i.e., the length of the carbon fiber bundle, significantly influences the morphological characteristics of the carbon cloth GDL. This aspect has rarely been reported in previous studies. To investigate the effect of the *LTR* on the transport properties of carbon cloth GDLs, 300 samples with a thickness of 200 μm, a porosity of 0.78, a fiber diameter of 7 μm, and random lengths ranging from 100 to 1000 μm were reconstructed, corresponding to an *LTR* of 0.5 to 5. The carbon cloth GDLs with different *LTRs* are shown in Figure 6. An increase in length enlarges the elliptical region of the carbon fiber bundles, incorporating more carbon fibers into each bundle. As the *LTR* increases, the carbon cloth GDL becomes more expansive in morphology.

Figure 7 shows the transport properties of carbon cloth GDLs with different *LTRs*, where the pore characteristics change due to the increase in carbon fiber wavelength caused by the increase in length. Specifically, the contribution of the LTR to the transport properties reveals an interesting finding: as the length increases, effective diffusivity and permeability increase, and tortuosity decreases when the *LTR* ranges from 0.5 to 2. This indicates that within this range, increasing the length of the carbon fiber bundles can optimize gas transport pathways and enhance gas transport efficiency. After the *LTR* reaches 2, the effective diffusivity, permeability, and tortuosity tend to stabilize, and the discretized distribution of the computational results may be attributed to more subtle microstructural features.

## 4. Conclusions

In this study, a stochastic reconstruction model based on the real structural parameters of carbon cloth GDLs was established, and a random walk algorithm was developed to analyze the effects of different porosities, carbon fiber diameters, and *LTRs* on the transport properties of carbon cloth GDLs. The main conclusions are as follows:(1)The impact of porosity on carbon cloth GDL transport properties was examined in this study. It was found that as porosity increased, the number of carbon fibers decreased, thereby enhancing gas and liquid transport while reducing mechanical strength. A modified Kozeny–Carman relation was utilized to improve the accuracy of permeability predictions, facilitating better modeling of fuel cell performance.(2)The pore structure and gas permeability of carbon cloth GDLs were significantly impacted by the diameter of carbon fibers. In this study, 300 samples with a constant porosity but varying fiber diameters were generated, revealing that larger diameters result in a reduced fiber count while maintaining the structural characteristics. Effective diffusivity, permeability, and tortuosity were analyzed, demonstrating strong correlations with fiber diameter. The modified relations that account for fiber diameter provided enhanced predictive accuracy for GDL transport properties.(3)The morphological characteristics of the carbon cloth GDLs were significantly influenced by the length of the carbon fiber bundles. In this study, 300 samples with varying *LTRs* were reconstructed, revealing that increasing length enhances gas transport efficiency up to an LTR of 2. Beyond this point, the properties stabilize, and the discrete distribution of computational results may be attributed to more subtle microstructural features.

## Figures and Tables

**Figure 1 nanomaterials-15-00259-f001:**
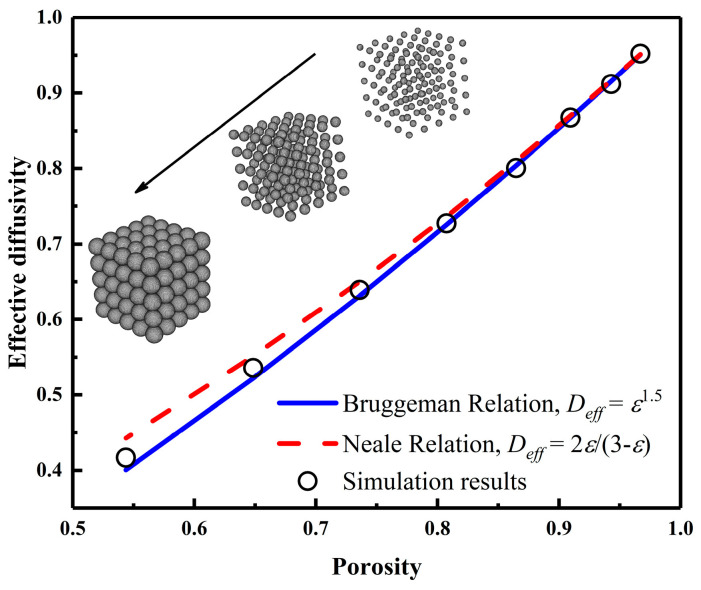
Comparison of the results of the random walk model for calculating effective diffusivity with previous relations.

**Figure 2 nanomaterials-15-00259-f002:**
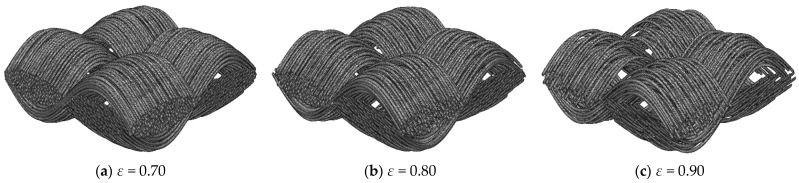
The three-dimensional structures of carbon cloth GDLs with different porosities.

**Figure 3 nanomaterials-15-00259-f003:**
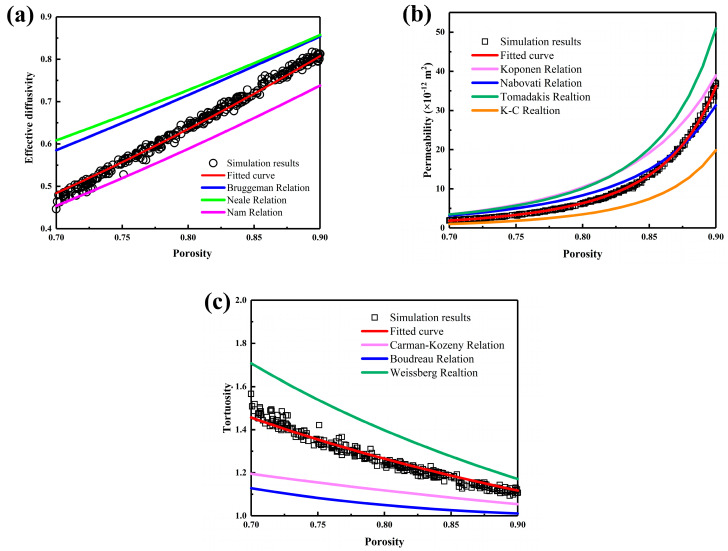
The transport properties of carbon cloth GDLs with different porosities: (**a**) effective diffusivity, (**b**) permeability, and (**c**) tortuosity.

**Figure 4 nanomaterials-15-00259-f004:**
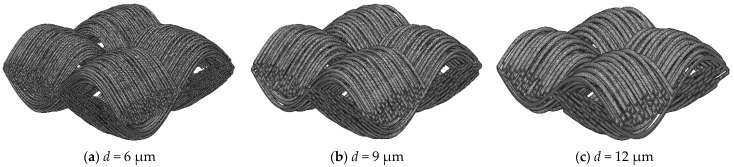
The three-dimensional structures of carbon cloth GDLs with different fiber diameters.

**Figure 5 nanomaterials-15-00259-f005:**
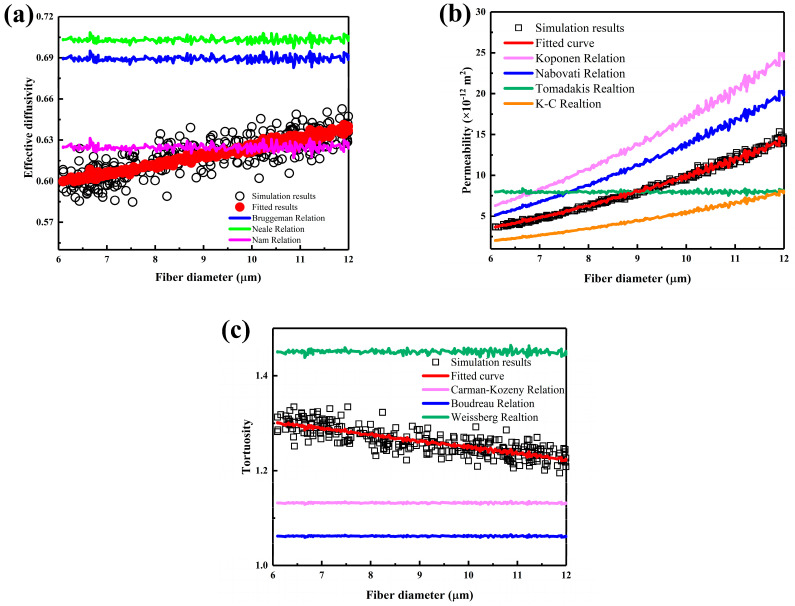
The transport properties of carbon cloth GDLs with different fiber diameters: (**a**) effective diffusivity, (**b**) permeability, and (**c**) tortuosity.

**Figure 6 nanomaterials-15-00259-f006:**
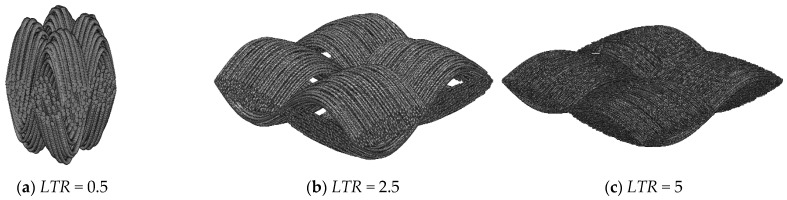
Three-dimensional structures of carbon cloth GDLs with different fiber diameters.

**Figure 7 nanomaterials-15-00259-f007:**
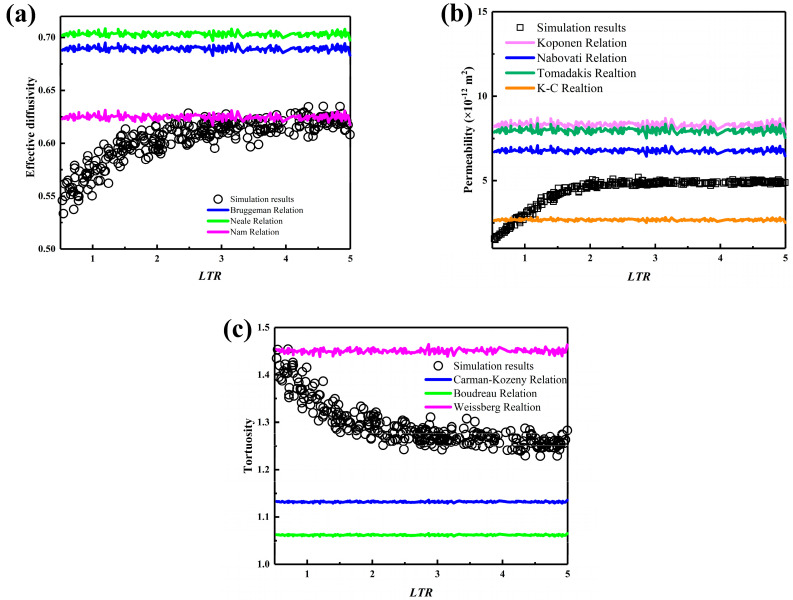
The transport properties of carbon cloth GDLs with different LTRs: (**a**) effective diffusivity, (**b**) permeability, and (**c**) tortuosity.

**Table 1 nanomaterials-15-00259-t001:** The relations of transport properties proposed by previous researchers.

Author	Parameter	Relation
Bruggeman et al. [22]	Effective diffusivity	$D_{eff}=\varepsilon^{1.5}$
Neale et al. [23]	Effective diffusivity	$D_{eff}=\frac{2\varepsilon }{3-\varepsilon }$
Nam et al. [25]	Effective diffusivity	$D_{eff}=\varepsilon {\left(\frac{\varepsilon -0.11}{0.89}\right)}^{0.785}$
Nabovati et al. [26]	Permeability	$K=0.491r^{2}{\left(\sqrt{\frac{0.9257}{1-\varepsilon }}-1\right)}^{2.31}$
Koponen et al. [27]	Permeability	$K=\frac{5.55r^{2}}{\exp\left[10.1\left(1-\varepsilon \right)\right]-1}$
Tomadakis et al. [28]	Permeability	$K=6.2805\times 10^{-13}\frac{\varepsilon }{{\left(\ln\varepsilon \right)}^{2}}$
Kozeny–Carman Relation	Permeability	$K=\frac{d^{2}}{180}\frac{\varepsilon^{3}}{{\left(1-\varepsilon \right)}^{2}}$
Carman [29]	Tortuosity	$\tau =\varepsilon^{-0.5}$
Boudreau et al. [30]	Tortuosity	$\tau =1+\frac{{\left(1-\varepsilon \right)}^{2}}{\varepsilon }$
Weissberg et al. [31]	Tortuosity	$\tau =\varepsilon^{-1.5}$

## Data Availability

Data is contained within the article.

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
