# Peer review of "Structural Parameters on the Effective Transport Properties of Carbon Cloth Gas Diffusion Layers: Random Walk Simulations"

_nanomaterials, 2025, doi:10.3390/nano15040259_

Round 1

Reviewer 1 Report

Comments and Suggestions for Authors

The review deals with research topics at the forefront of research, such as the development of new materials for fuel cells. The paper is well written. The methodology to material synthesis characterizations and simulations are correct. The presentation is clear enough and the discussion of the results is coherent. 

I have a recommendation for authors. You can include an interesting result from a work in lines 34, 47 or 58 as a reference. It should be: “the effect of the electrode permeability on the PEMFC performance depends strongly upon the flow-field pattern”. The cites is the following: https://doi.org/10.1016/S0378-7753(03)00081-8

In other hand, in line 257, a cite is missing.

In my opinion, the manuscript should be accepted with minor changes after correcting the text.

Author Response

#Reviewer1,

The review deals with research topics at the forefront of research, such as the development of new materials for fuel cells. The paper is well written. The methodology to material synthesis characterizations and simulations are correct. The presentation is clear enough and the discussion of the results is coherent.

Comments 1. I have a recommendation for authors. You can include an interesting result from a work in lines 34, 47 or 58 as a reference. It should be: “the effect of the electrode permeability on the PEMFC performance depends strongly upon the flow-field pattern”. The cites is the following: https://doi.org/10.1016/S0378-7753(03)00081-8

Response:

After evaluation, this study is considered crucial for readers to gain a more comprehensive understanding of the transport properties of porous media in electrodes. The citation has been added in line 37, thank you.

Comments 2. In other hand, in line 257, a cite is missing.

Response:

After verification, it was confirmed that a citation was indeed missing at this location, and it has now been corrected. Thank you.

Reviewer 2 Report

Comments and Suggestions for Authors

This simulation of carbon fibre membranes is well conceived, well constructed and well-presented.  Inevitably the model is an idealised construct, but has been ingeniously conceived to capture the essentiall parameters charcterising the real material.

The particular target identified by the authors is the application in fuel cells, but the reults of this study will be useful in any context where this type of membrane is employed.

Author Response

#Reviewer2,

This simulation of carbon fibre membranes is well conceived, well constructed and well-presented.  Inevitably the model is an idealised construct, but has been ingeniously conceived to capture the essentiall parameters charcterising the real material.

The particular target identified by the authors is the application in fuel cells, but the reults of this study will be useful in any context where this type of membrane is employed.

Response:

We are deeply grateful for the positive comments on our manuscript from Reviewer 2. As Reviewer 2 pointed out, this study offers a method for predicting the transport properties of porous media at low cost and with high efficiency. Previous studies typically focused on typical structures such as spheres and fibers. However, for porous media with different structural characteristics, previous empirical correlations lacked predictive accuracy. The prediction method for transport parameters proposed in this study is universally applicable, while the research on carbon cloth GDL is highly targeted.

Reviewer 3 Report

Comments and Suggestions for Authors

This study was conducted to clarify the relationship between the microstructure of carbon cloth GDL and its performance in PEM. The authors evaluated the effective diffusivity, permeability, and tortuosity when changing the structure of carbon cloth GDL and argued for the importance of porosity, fiber diameters, and LTD. In my opinion, the most significant achievement in this paper is the development of a stochastic reconstruction algorithm for carbon cloth GDLs. However, the results for porosity and fiber diameter from this simulation only show a general trend that could have been predicted without simulation. If possible, I recommend that the authors propose the best GDL structure including the detailed information of porosity, fiber diameters, and LTD based on this simulation.

The following points are my questions. Please address them before publishing the paper.

1.     Although the rules for random work are fairly simple, actual gas diffusion in GDL should be more complex. For instance, gas diffusion may show different behaviors depending on the target gases, such as hydrogen, oxygen, or water, due to the correlation between the carbon surface and these gases. Please provide the appropriate reason why authors can ignore these correlations and choose this simple model.

2.     Aothurs mentioned, "As the porosity increases, the number of carbon fibers in the carbon fiber bundles decreases, and the carbon cloth GDL becomes more loose, which will be more favorable for gas and liquid water transport, but with worse mechanical properties"

However, they didn't investigate the mechanical properties. This claim is not based on the results of this research but just on general ideas that the authors believe. It may mislead the readers.

3.     Although the authors compare various models in Figures 3, 5, and 7, they hardly explain the physical meaning of each model. Please clarify this point and take it into consideration when discussing the simulation results of this study.

4.     Why did the authors choose a porosity of 0.78?

5.     Authors should mention the fiber diameter in Figures 6 and 7.

Minor improvements

L257: Nabovati relation]→Nabovati relation

And add a reference for Nabovati relation

L328: LTR→Length-to-Thickness Ratio (LTR)

Author Response

#Reviewer3,

This study was conducted to clarify the relationship between the microstructure of carbon cloth GDL and its performance in PEM. The authors evaluated the effective diffusivity, permeability, and tortuosity when changing the structure of carbon cloth GDL and argued for the importance of porosity, fiber diameters, and LTD. In my opinion, the most significant achievement in this paper is the development of a stochastic reconstruction algorithm for carbon cloth GDLs. However, the results for porosity and fiber diameter from this simulation only show a general trend that could have been predicted without simulation. If possible, I recommend that the authors propose the best GDL structure including the detailed information of porosity, fiber diameters, and LTD based on this simulation.

The following points are my questions. Please address them before publishing the paper.

Comments 1. Although the rules for random work are fairly simple, actual gas diffusion in GDL should be more complex. For instance, gas diffusion may show different behaviors depending on the target gases, such as hydrogen, oxygen, or water, due to the correlation between the carbon surface and these gases. Please provide the appropriate reason why authors can ignore these correlations and choose this simple model.

Response:

Based on the previous studies on predicting the transport properties of porous media using the random walk method, the assumptions made in those studies have been summarized and declared in the manuscript.

In this study, the following widely adopted assumptions need to be declared: (1) The transition rules are equiprobable, allowing particles to randomly select directions to move within a three-dimensional space; (2) The interactions between the gas and the medium surface, as well as the interactions among gas molecules, are neglected, and only the molecular diffusion in free space is considered.

Comments 2. Aothurs mentioned, "As the porosity increases, the number of carbon fibers in the carbon fiber bundles decreases, and the carbon cloth GDL becomes more loose, which will be more favorable for gas and liquid water transport, but with worse mechanical properties" However, they didn't investigate the mechanical properties. This claim is not based on the results of this research but just on general ideas that the authors believe. It may mislead the readers.

Response:

References have been added to support this viewpoint at this statement.

Comments 3. Although the authors compare various models in Figures 3, 5, and 7, they hardly explain the physical meaning of each model. Please clarify this point and take it into consideration when discussing the simulation results of this study.

Response:

The effective diffusivity represents the ratio of the actual diffusion rate of a gas or liquid in a porous medium to the diffusion rate in an ideal (non-porous medium or open space). The permeability describes the ability of a porous medium to permit the passage of a liquid or gas, reflecting the connectivity of the pore structure and the size of the pores. The tortuosity indicates the complexity of the path that a fluid or gas takes through a porous medium relative to the shortest straight-line path. The higher the tortuosity, the more tortuous the path that the fluid or gas takes through the medium. The physical meanings of these transport parameters are well known.

Regarding the content presented in Figures 3, 5, and 7, they have been given respectively. By changing the settings of the random reconstruction model parameters, 300 different carbon cloth GDL physical structures with varying porosities, fiber diameters, and LTR were obtained, and the predicted transport properties are shown in Figures 3, 5, and 7. Concerning the physical significance of the model, each curve represents a predicted correlation model for transport properties obtained by controlling different parameters such as porosity and carbon fiber diameter. The above content has been declared in the manuscript.

Comments 4. Why did the authors choose a porosity of 0.78?

Response:

A porosity of 0.78 is a typical value for fuel cell GDLs. 

  • R. Molaeimanesh, M. Dahmardeh, Pore‐scale analysis of a PEM fuel cell cathode including carbon cloth gas diffusion layer by lattice Boltzmann method, Fuel Cells. 21 (2021) 208–220. https://doi.org/10.1002/fuce.202000191.
  • Luo, Y. Ji, Chao Yang Wang, P.K. Sinha, Modeling liquid water transport in gas diffusion layers by topologically equivalent pore network, Electrochimica Acta. 55 (2010) 5332–5341. https://doi.org/10.1016/j.electacta.2010.04.078.

Comments 5. Authors should mention the fiber diameter in Figures 6 and 7.

Response:

The carbon fiber diameter of 7 μm has been declared in the manuscript. 

Comments 6. L257: Nabovati relation]→Nabovati relation. And add a reference for Nabovati relation. L328: LTR→Length-to-Thickness Ratio (LTR)

Response:

The format error has been corrected, and the citation has been supplemented.